# Computation-guided asymmetric total syntheses of resveratrol dimers

Masaya Nakajima [1✉], Yusuke Adachi[1] & Tetsuhiro Nemoto [1✉]

Although computational simulation-based natural product syntheses are in their initial stages of development, this concept can potentially become an indispensable resource in the field of organic synthesis. Herein we report the asymmetric total syntheses of several resveratrol dimers based on a comprehensive computational simulation of their biosynthetic pathways. Density functional theory (DFT) calculations suggested inconsistencies in the biosynthesis of vaticahainol A and B that predicted the requirement of structural corrections of these natural products. According to the computational predictions, total syntheses were examined and the correct structures of vaticahainol A and B were confirmed. The established synthetic route was applied to the asymmetric total synthesis of (−)-malibatol A, (−)-vaticahainol B, (+)-vaticahainol A, (+)-vaticahainol C, and (−)-albiraminol B, which provided new insight into the biosynthetic pathway of resveratrol dimers. This study demonstrated that computation-guided organic synthesis can be a powerful strategy to advance the chemical research of natural products.

[1] Graduate School of Pharmaceutical Sciences, Chiba University, 1-8-1, Inohana, Chuo-ku, Chiba 260-8675, Japan. ✉email: m.nakajima@chiba-u.jp; tnemoto@faculty.chiba-u.jp

riven by the rapid advances in computer performance, the role of theoretical calculations in the field of organic chemistry has changed considerably over the past few decades. Density functional theory (DFT) calculations can be used to simulate and predict various molecular properties in a virtual environment, which is now recognized as one of the most important technologies to elucidate reaction mechanisms and design useful functional molecules. Computational programs that can simulate ideas for complex molecule synthesis and verify the validity of synthetic plans provide a precise guide for conducting efficient experiments in real laboratory settings. Although such techniques are in their initial stages of development, they will potentially become indispensable resources in the field of organic synthesis[1–3]. Computational chemistry also plays an important role in the field of biosynthetic chemistry[4,5]. Computational insight into the biosynthetic mechanisms of natural products can help us understand their metabolic pathways, as well as verify the structures of biosynthetic intermediates. Furthermore, computational simulations of biosynthetic pathways can provide a logical guide and platform for the biomimetic total syntheses of natural products[6]. Therefore, computational chemistry has gained immense attention from organic chemists and is currently applied to various fields of chemical sciences.

Resveratrol, the well-known causative agent of the "French paradox"[7,8], and related resveratrol oligomers (ROs) are polyphenols found in a variety of plant species. ROs are divergently biosynthesized starting from regioselective coupling reactions of radical species produced by one-electron oxidation of resveratrol and over 200 ROs have been isolated from natural resources. This class of polyphenol natural products exhibits a wide range of bioactivities, such as antioxidant, antitumor, and cardiovascular activities, highlighting their potential utility in drug discovery research[9,10]. The pharmacological profiles and structural complexity of ROs have made them attractive targets for synthetic organic chemists[11–17]. Owing to their structural complexity deriving from the diverse oligomerization patterns and large number of stereocenters, as well as the low amounts isolable from plant resources, considerable effort is required to determine the structures by natural product chemistry research techniques. Computational simulation of the biosynthetic pathways can be an effective alternative to solve this issue. Biosynthetic pathways of ROs, even for the simplest resveratrol dimers, have not been comprehensively studied using computational simulation techniques. This background prompted us to perform DFT calculations of the biosynthetic pathways for various resveratrol dimers starting from resveratrol.

All possible natural and non-natural diastereomers were comprehensively simulated using DFT calculation to understand the biosynthesis from resveratrol to hopeanol (Fig. 1). Dimerization of resveratrol (1) resulted in the production of ε-viniferin (2) (see Supplementary Figs. 7 and 8), which could be the common precursor of balanocarpol[18], ampelopsin A[19], hemsleyanol A[20], and acuminatol (4)[21], via epoxidation and subsequent Friedel–Crafts-type cyclization reactions (see Supplementary Figs. 9 to 12). Baranocarpol or acuminatol could be transformed to malibatol A (5)[22] (absolute configuration has not been determined), which was computationally simulated as the biosynthetic intermediate for vaticahainol B (6) and vaticahainol A (7)[23]. Finally, transformation to hopeahainol A (8)[24] and hopeanol (9)[24] were calculated from vaticahainol A (see Supplementary Figs. 13 and 14). The obtained potential energy diagrams revealed inconsistencies in the energy profiles of the biosynthetic process of vaticahainol A and B. Generation of the C7b epimer of vaticahainol B (epi-6) and the C8b epimer of vaticahainol A (epi-7) was favorable. The biosynthesis of hopeahainol A from C8b-epi-vaticahainol A was also energetically feasible. These

computational inconsistencies led us to hypothesize that structural revisions might be required for vaticahainol A and B.

Herein, we report the computation-guided total synthesis and structural correction of vaticahainol A and B based on comprehensive DFT calculations of the biosynthetic pathways. Asymmetric total syntheses of the (+)- and (−)-enantiomers of malibatol A and vaticahainol A, B, and C (albiraminol B) are also achieved by late-stage kinetic resolution using an enantioselective acylation of a racemic malibatol A derivative.

## Results

**DFT calculation of biosynthetic pathway.** Structure optimizations were carried out with the Gaussian 16 program[25] at 298.15 K, using the ωB97X-D[26] functional with an ultrafine grid and the LANL2DZ[27–29] (for Fe) and 6-31 G(d,p) (for other atoms) basis sets, which were found to be optimal in our benchmark study comparing the results with ab initio calculations at the MP2 level (see Supplementary Fig. 4). All the biosynthetic calculations were performed in water solvent using the solvation model based on density (SMD)[30]. The other calculations were performed in gas phase. Harmonic vibrational frequencies were computed with the same level of theory to confirm no imaginary vibration was observed for the optimized structure, and only one imaginary vibration was observed for the transition state. The intrinsic reaction coordinate (IRC)[31] method was used to track minimum energy paths from transition structures to the corresponding local minima. The conformation search was performed by the Monte Carlo method with the Spartan18 program[32] using an energy within 25 kcal/mol and an upper limit of 1000 conformers at the MMFF level. All the calculated configurations were subjected to structural optimization and vibrational frequency calculation at the DFT level (ωB97X-D/6-31 G(d, p) in water (SMD)) using the Gaussian16 program, and the global minimum was determined.

For biosynthesis from malibatol A (5) to vaticahainol B (6), we simulated a multi-step pathway, including epoxidation using compound I (cpd I), which was a model structure of the active unit of cytochrome P450[33–35] (Fig. 2a). Before calculating the biosynthetic pathway, the global minimum (SM) of malibatol A was calculated by a conformational search (see Supplementary Fig. 1). We attempted to calculate the transition state for epoxidation by cpd I based on the calculated global minimum structure. For the biosynthesis of vaticahainol B, the epoxidation must proceed from the same side as the Ar group on C7a. Due to the steric hindrance of the pseudo-axial Ar group on C7a in **SM**, however, the global minimum of malibatol A, the bulky cpd I, could not be approached. Therefore, no transition state for this approach mode to **SM** was found. Instead, the desired epoxidation transition state (**TS1**) was found from the local minimum **SM2**, where the seven-membered ring of the global minimum flips and the Ar group of C7a is in the pseudo-equatorial position (blue energy diagram). On the other hand, a transition state in which epoxidation proceeds from the opposite side of the pseudo-axial Ar group on C7a, was found from the global minimum (**SM**). This transition state yields the C7b epimer of vaticahainol B, which has not been isolated as a natural product. Because of the conformational difference, the activation barrier of **TS1** was 8.7 kcal/mol higher than that of **TS1′**, indicating that the biosynthesis of C7b-epi-vaticahainol B was kinetically favored. The reactions with the quartet state of iron were also calculated and showed a similar trend (see Supplementary Fig. 15). We also considered epoxidation under a non-enzymatic environment: epoxidation with hydroperoxyl radicals. Because hydroperoxyl radicals are very small molecules, both transition states were found from the global minimum conformation (**SM**).

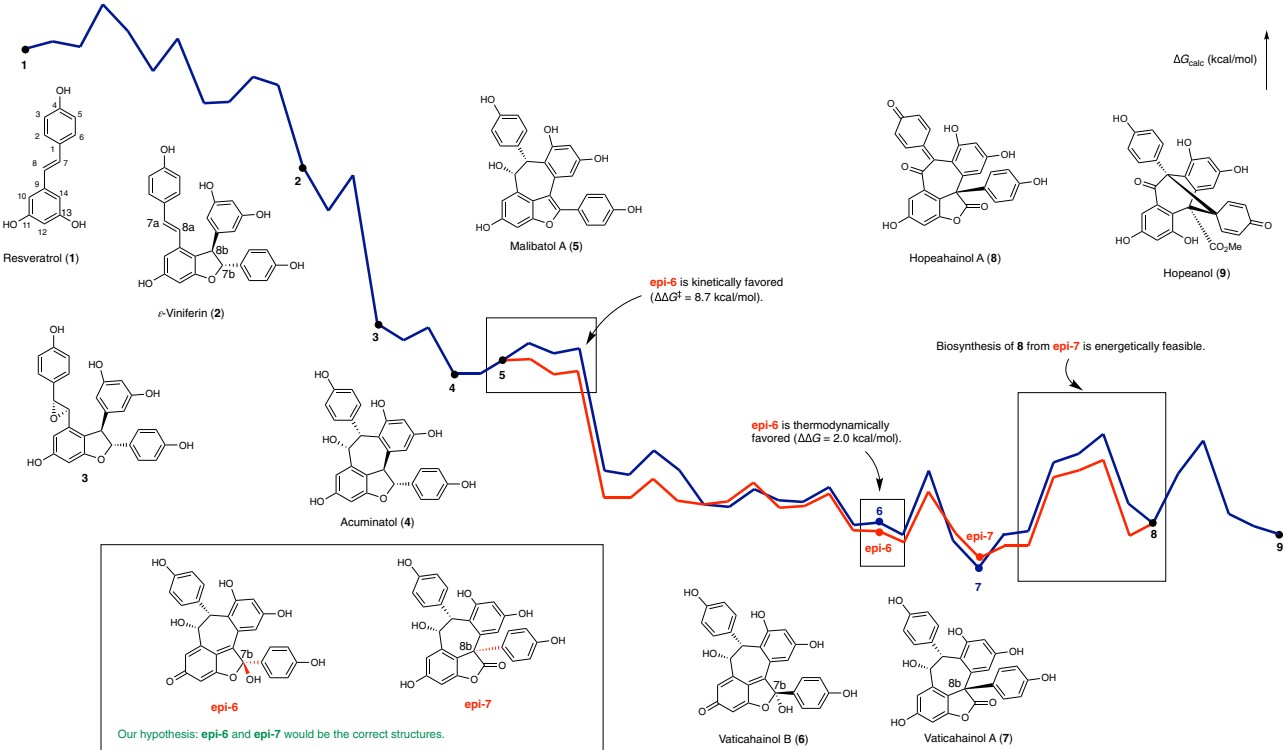

**Fig. 1 Summary of the results obtained from DFT calculations.** Energy diagram of the biosynthetic route of resveratrol dimers and the predicted correct structures of vaticahainol B and A.

Hydroperoxyl radicals, however, preferred to approach from the opposite side of the C7a Ar group (see Supplementary Fig. 16). Elimination (**TS2**) of the bulky iron(III) porphyrin allows for structural transformation of the unstable (Ar of C7a: equatorial) epoxide (**INT2**) to the stable (Ar of C7a: axial) conformation (**Epoxy 1**). Subsequent epoxide-opening (**TS3** and **TS3′**) and proton shuttling by water molecules (**TS4** and **TS4′**) were calculated to proceed at similarly low activation energies for both routes.

It was also observed that the energy of C7b-epi-vaticahainol B (**PD1′ (epi-6)**) was 2.0 kcal/mol lower than that of vaticahainol B (**PD1 (6)**) (see Supplementary Fig. 2 for the conformation search of **PD1** and **PD1′**). The simulated biosynthetic process revealed that C7b-epi-vaticahainol B (**PD1′(epi-6)**) was the kinetically and thermodynamically favored product.

Next, the biosynthetic pathways for the formation of vaticahainol A (**7**) and its C8b epimer (**epi-7**) were computationally simulated (Fig. 2b). Under neutral conditions, the activation energy of the rearrangement process from vaticahainol B (**PD1(6)**) to A (**PD2(7)**) was >40 kcal/mol (see Supplementary Fig. 19), while the activation energy of the reaction resulting in the formation of **TS5** under Brønsted acidic conditions was 27.2 kcal/mol. The activation energy of the rearrangement reaction (**TS5′**) to form C8b-epi-vaticahainol A (**PD2′(epi-7)**) was 8.5 kcal/mol lower than that of the reaction (**TS5**) to form vaticahainol A (**PD2 (7)**). The free energy of vaticahainol A (**PD2 (7)**) was 3.1 kcal/mol lower than that of C8b-epi-vaticahainol A (**PD2′ (epi-7)**), indicating that the formation of vaticahainol A (**PD2 (7)**) was thermodynamically favored (see Supplementary Fig. 3 for the conformation search of **PD2** and **PD2′**). If there was an epimerization pathway from C7b-epi-vaticahainol B to vaticahainol B, vaticahainol A would be thermodynamically generated. Therefore, we attempted to identify the interconversion pathway between C7b-epi-vaticahainol B (**PD1′(epi-6)**) and vaticahainol B (**PD1 (6)**). Hemiacetal opening steps to form the

diketone intermediates via **TS6** or **TS6′** and the subsequent steps of C–C bond rotation in the aryl ketone unit via **TS7** were calculated; the energy required for this interconversion process was >30 kcal/mol. This suggested that epimerization rarely occurred to form vaticahainol B (**PD1 (6)**) from C7b-epi-vaticahainol B (**PD1′(epi-6)**). Moreover, we verified **TS1**, **TS1′**, and **TS7** with various functional/basis-set, and all calculation results supported that the generation of C7b-epi-vaticahainol B was kinetically favored and its epimerization hardly proceeded (see Supplementary Fig. 17). From these results, we hypothesized that the correct structures of vaticahainol B and A were the C7b (**epi-6**) and C8b (**epi-7**) epimers of the reported structures **6** and **7**, respectively. We thus attempted the total syntheses of the identified **epi-6** and **epi-7** according to the computational prediction.

**Total syntheses of resveratrol dimers.** We first prepared racemic Ac-Me5-malibatol A (**10**) starting from resveratrol (**1**) over nine steps using the reported protocols[36–38] (Fig. 3a). The simulated activation energies indicated that once the benzofuran moiety in malibatol A is epoxidized, the epoxide opening reaction should spontaneously occur at room temperature to form vaticahainol B (Fig. 2a; **INT2**, **2′** to **PD1**, **1′**). Thus, after acetylation of **10**, we examined the epoxidation of **11** in the presence of widely used epoxidizing agents such as *m*-chloroperoxybenzoic acid or dimethyl-dioxirane. A complex mixture of the products was obtained upon completion of the reaction, however, probably because of over-oxidation. In contrast, reaction of compound **11** with 2 equiv of cerium (IV) ammonium nitrate (CAN) generated the carbocation at the C7b position, which was then trapped with $H_2O$ to give the desired dearomatized product **12** in 79% yield as a single diastereomer. Comparing the calculated transition states between **TS8** and **TS8′**, the energy required for the addition of the water molecules above the plane was 3.4 kcal/mol lower than that required for the addition of the water molecules below the plane,

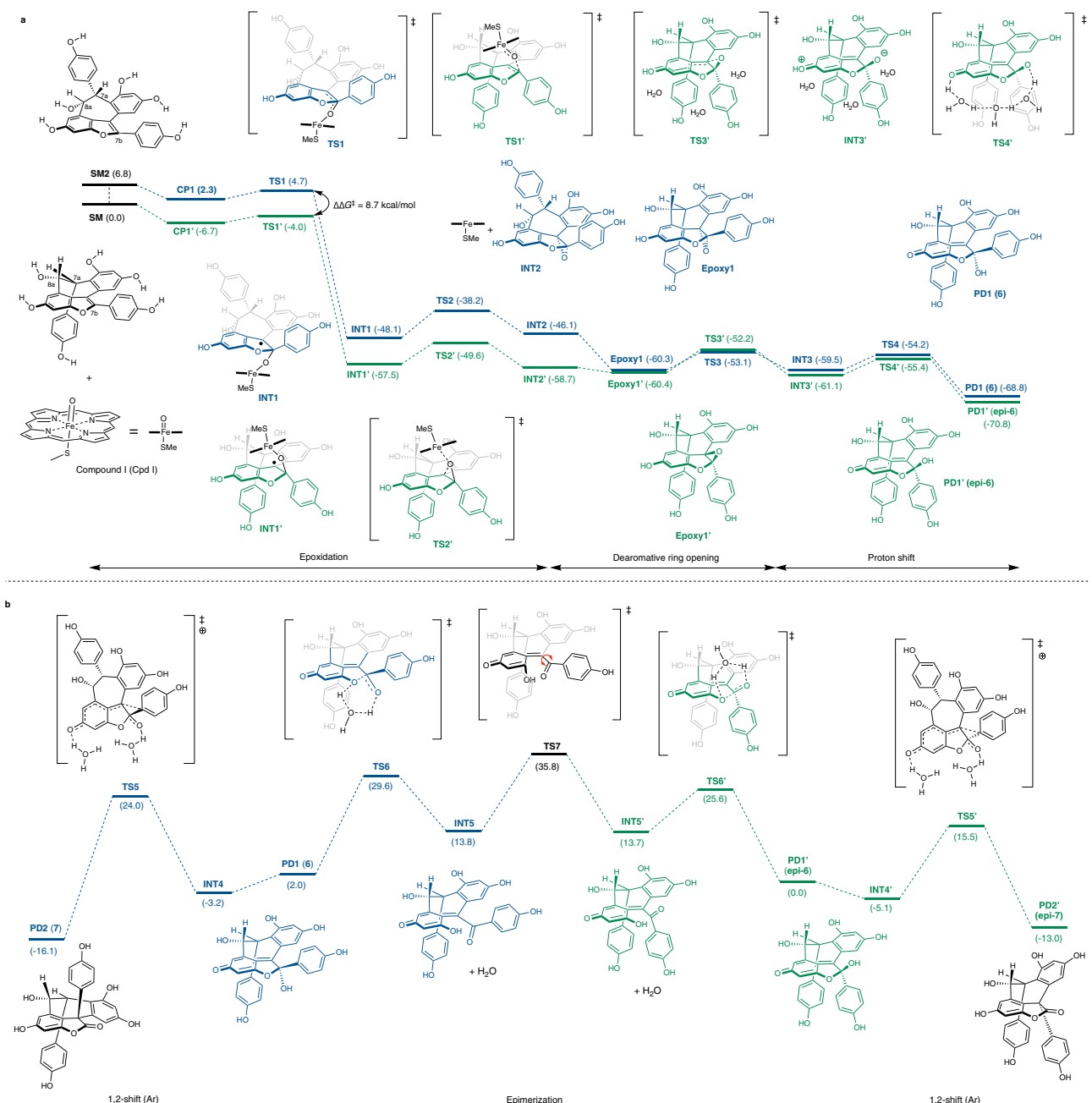

**Fig. 2 DFT calculation of biosynthetic pathway of vaticahainol A and B. a** Energy diagram for the biosynthesis of vaticahainol B (**6**) and its epimer (**epi-6**). **b** Energy diagram for the biosynthesis of vaticahainol A (**7**) and its epimer (**epi-7**). SM starting material, CP complex, TS transition state, INT intermediate, PD product.

indicating that the relative stereochemistry of the product would be C7b-epi-Ac-Me₄-vaticahainol B (see Supplementary Fig. 20). Sequential deprotection of benzylic alcohol and four phenols yielded the product in 76% yield in 2 steps, and the spectroscopic data of this compound matched those for vaticahainol B in the literature[23]. The product structure was unequivocally determined by X-ray crystal structure analysis, revealing that the product was C7b-epi-vaticahainol B (**epi-6**), which was in accordance with our computational prediction. A subsequent 1,2-aryl shift was next attempted under acidic conditions to synthesize vaticahainol A (**7** or **epi-7**). Treatment of **epi-6** with TFA at 70 °C gave the rearranged product in 62% yield. The spectroscopic data of this compound also matched those for vaticahainol A in the literature[23]. To determine its relative stereochemistry, all phenolic

hydroxyl groups were methylated to afford compound **14**, which could be also synthesized from **12** through the 3-step sequence involving an acid-promoted 1,2-aryl shift and subsequent protecting group manipulations of compound **13**. The structure of **13** was determined by X-ray crystal structure analysis, which revealed a syn relationship between the two aromatic rings at the C7a and C8b positions. Thus, it was concluded that the C7b-epimer (**epi-7**) represented the correct structure of vaticahainol A. Furthermore, oxidation of **14** using 2-iodoxybenzoic acid (IBX), followed by demethylation with BBr₃, resulted in the total synthesis of hopeahainol A (**8**)[24] (18% yield, 2 steps) and a formal total synthesis of hopeanol (**9**)[13,24]. Nicolaou and his co-workers reported that the reactivity of this IBX oxidation process was significantly affected by the diastereomeric substrate structure:

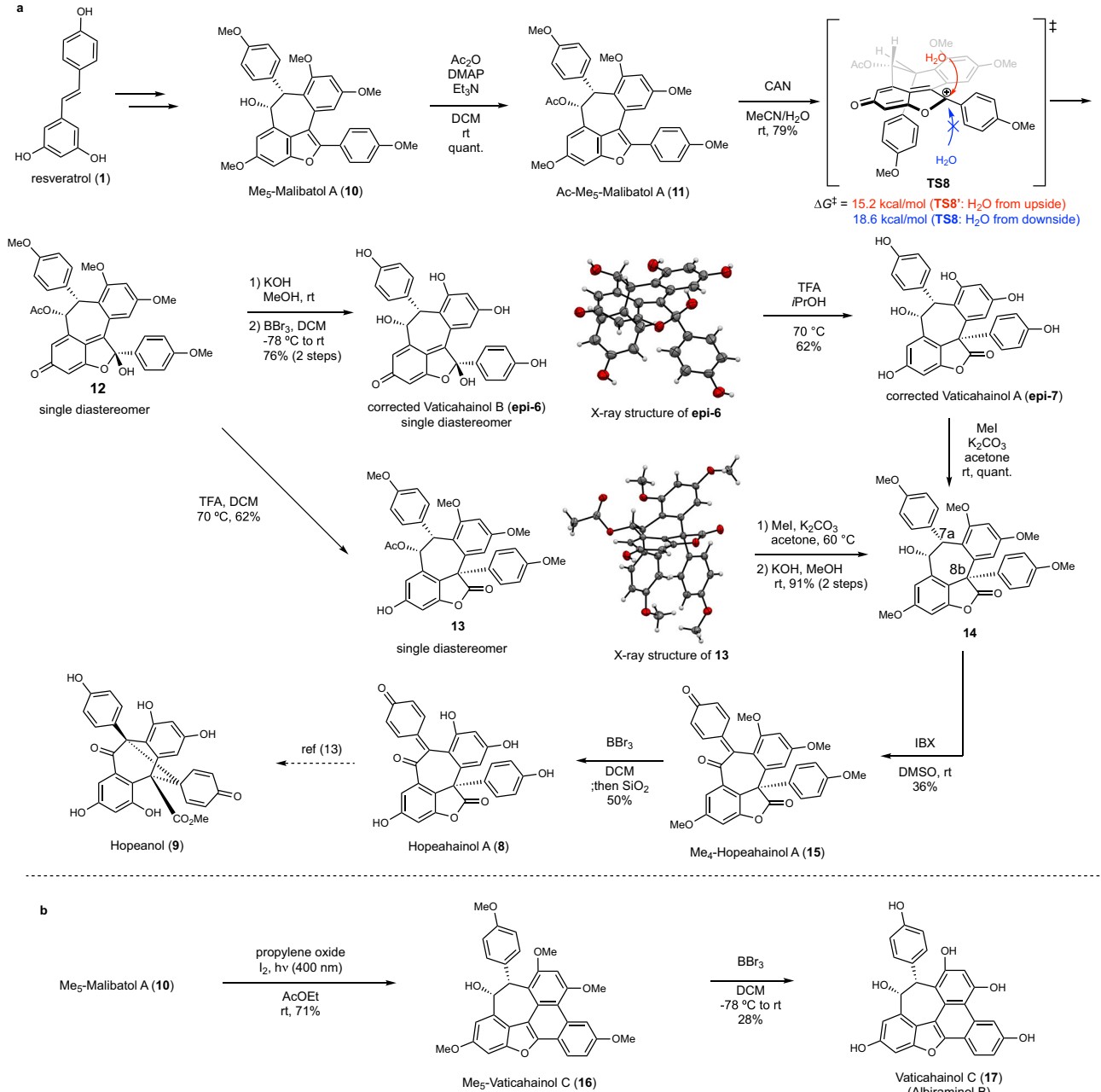

**Fig. 3 Total syntheses of resveratrol dimers. a** Total syntheses of corrected vaticahainol A (**epi-7**), vaticahainol B (**epi-6**), and hopeahainol A. **b** Total syntheses of vaticahainol C (albiraminol B). DMAP 4-dimethylaminopyridine, DCM dichloromethane, CAN cerium (IV) ammonium nitrate, TFA trifluoroacetic acid, IBX 2-iodoxybenzoic acid, DMSO dimethyl sulfoxide.

dearomative oxidation of the C7a aryl group did not occur using a substrate with an anti-stereochemical relationship between the two aromatic rings at the C7a and C8b positions, although smooth oxidation occurred with the use of a syn-type substrate[13]. This reactivity can be predicted from the simulated energy diagram in Fig. 1, where the epi-**7** to **8** conversion proceeded with a lower energy barrier than that for **7** to **8** (see Supplementary Fig. 13). Therefore, it was synthetically and computationally suggested that the correctly identified vaticahainol A (**epi-7**) was a biosynthetic intermediate of hopeahainol A (**8**) and hopeanol (**9**). Total synthesis of vaticahainol C (**17**)[23] (albiraminol B[39]) was also achieved using a purple (400 nm) light-induced biomimetic 6π-electron cyclic reaction[40] of **10** as the key step (Fig. 3b) (see Supplementary Fig. 18 for the calculation of its biosynthesis).

**Optimization of kinetic resolution system.** After establishing a synthetic route for the correct structures of vaticahainol A and B, we next focused on the asymmetric total synthesis of a series of resveratrol dimers to determine the absolute stereochemistry. We chose kinetic resolution (KR) of racemic Me5-malibatol A (**10**) through enantioselective acylation of benzylic alcohol to obtain both enantiomers of the intermediate. We first performed computational screening of easily available chiral acylating agents using the DFT method to determine a suitable KR system for compound **10** (Fig. 4). Although the transition states of acylation were highly charged, we expected that the ΔΔG derived from steric hindrance of the asymmetric catalyst even in the gas phase would be a guideline for choosing an asymmetric catalyst. For efficient KR, a bulky anhydride such as isobutyric anhydride is often used. We selected acetic anhydride to compare the several

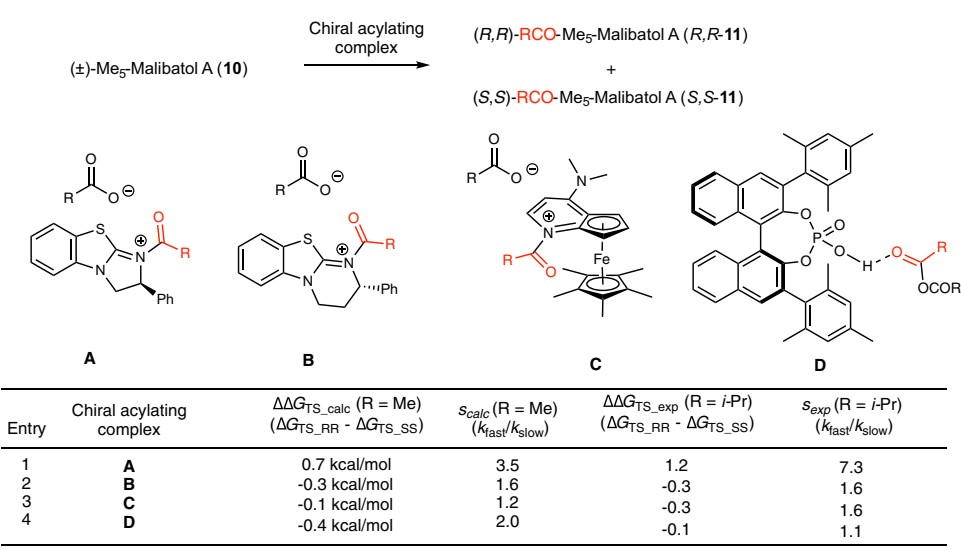

| Entry | Chiral acylating complex | $\Delta\Delta G_{TS\_calc}$ (R = Me) ($\Delta G_{TS\_RR} - \Delta G_{TS\_SS}$) | $s_{calc}$ (R = Me) ($k_{fast}/k_{slow}$) | $\Delta\Delta G_{TS\_exp}$ (R = i-Pr) ($\Delta G_{TS\_RR} - \Delta G_{TS\_SS}$) | $s_{exp}$ (R = i-Pr) ($k_{fast}/k_{slow}$) |
|---|---|---|---|---|---|
| 1 | A | 0.7 kcal/mol | 3.5 | 1.2 | 7.3 |
| 2 | B | -0.3 kcal/mol | 1.6 | -0.3 | 1.6 |
| 3 | C | -0.1 kcal/mol | 1.2 | -0.3 | 1.6 |
| 4 | D | -0.4 kcal/mol | 2.0 | -0.1 | 1.1 |

**Fig. 4 DFT calculations of kinetic resolution.** Scope of kinetic resolution system and experimental validation for the calculation results.

chiral acylating reagents, however, by considering the number of conformers and calculation cost. Then, 6–7 TSs with different conformations were calculated in the presence of the chiral acetylating complex (**A–D**)[41–44] to afford (S,S)-**11** and (R,R)-**11** (see Supplementary Fig. 21 for the energy of each conformer). The Boltzmann distribution was considered while calculating the $\Delta\Delta G_{TS}$ ($\Delta G_{TS\_RR} - \Delta G_{TS\_SS}$) values. The maximum value of $\Delta\Delta G_{TS}$ was calculated to be 0.7 kcal/mol in the presence of Birman's base **A**, and selective formation of (S,S)-**11** was predicted when using Birman's base with an S configuration. An initial experimental KR study was attempted to confirm whether the calculated tendency correlated with the experimental results. When we used acetic anhydride (R = Me), the KR was not successful with any chiral reagent due to background reactions. The use of isobutyric anhydride (R = iPr) prevented the background reaction to yield the s value: chiral acylating complex **A** showed the best $\Delta\Delta G_{TS}$ and s value (1.2 kcal/mol and 7.3, respectively), and **B-D** showed a small $\Delta\Delta G_{TS}$ and s value (−0.1 to −0.3 kcal/mol and 1.1 to 1.6, respectively), suggesting that **A** would be the best reagent for this KR as predicted by model calculation.

**Kinetic resolution**. Thus, we attempted to optimize the reaction conditions using Birman's base (S)-**24** (Fig. 5a). After several attempts, we successfully obtained (−)-Me₅-malibatol A ((−)-**10**; s = 7.7; $\Delta G^{\ddagger} = 1.2$; isolated yield: 35%; 95% ee). Furthermore, (+)-Me₅-malibatol A ((+)-**10**; −40% ee), which was obtained by hydrolysis of (+)-iPrCO-Me₅-malibatol A (**25**), was further subjected to the kinetic resolution process in the presence of Birman's base (R)-**24**. (+)-Me₅-malibatol A ((+)-**10**) was successfully obtained in 55% and −97% ee. Because KR was experimentally successful with isobutyric anhydride, we performed the calculation again using complex **A** (R = iPr). Due to the rotational isomers of the iPr group, we considered 115 conformers (see Supplementary Fig. 22 for the energy of each conformer) and obtained s = 10.2 and $\Delta G^{\ddagger} = 1.4$ kcal/mol, which correlated with the experimental results (s = 7.7 and $\Delta G^{\ddagger} = 1.2$ kcal/mol). The absolute configurations of (−)-**10** and (+)-**10** were determined by X-ray crystal structure analysis after converting into (R,R)-Me₅-vaticahainol C ((−)-**16**) and (S,S)-Me₅-vaticahainol C ((+)-**16**) (Fig. 5b). These results validated our computational prediction.

**Asymmetric total syntheses of resveratrol dimers**. Using (−)-(R,R)-**10** and (+)-(S,S)-**10** with high optical purity,

asymmetric total syntheses of both enantiomers of malibatol A and vaticahainol A, B, and C (albiraminol B) were achieved (Fig. 5b). The optical rotation and circular dichroism spectral profiles of the synthetic and isolated natural products were compared, revealing that (−)-malibatol A, (−)-vaticahainol B, (+)-vaticahainol A, (+)-vaticahainol C, and (−)-albiraminol B were the natural enantiomers. Interestingly, (−)-albiraminol B and (+)-vaticahainol C, reported in the same relative configuration, were found to be enantiomers. The absolute configuration of (−)-albiraminol B was C7aR, C8aR, which is the same as that of natural (−)-malibatol A. In contrast, the absolute configuration of the natural vaticahainols was C7aS, C8aS, which is the same as that of non-natural (+)-malibatol A. Stereochemical correlations between the C7a and C8a positions suggested that (−)-(C7aR, C8aR)-acuminatol is the biosynthetic intermediate for (−)-albiraminol B and (−)-malibatol A (Fig. 5c). Similarly, (+)-(C7aS, C8aS)-balanocarpol, the C7a, C8a epimer of (−)-acuminatol, is suggested to be the biosynthetic intermediate for natural vaticahainols. Our experimental and computational results also revealed that natural (+)-vaticahainol A is the plausible biosynthetic intermediate of natural (+)-hopeahainol A and (+)-hopeanol. Although (+)-malibatol A has not yet been isolated as a natural product, its existence in nature is strongly suggested as the missing piece to complete the biosynthetic scheme of resveratrol dimers.

**Discussion**

We performed DFT calculations to simulate the biosynthetic pathway of resveratrol dimers, which suggested that the structures of vaticahainol A and B required corrections. Total synthesis according to the predicted biosynthetic pathway confirmed the correct structures of vaticahainol A and B. Furthermore, the kinetic resolution of (±)-Me₅-malibatiol A using Birman's base afforded both enantiomers of this intermediate with high optical purity, which were successfully applied to the asymmetric total synthesis of (−)-malibatol A, (−)-vaticahainol B, (+)-vaticahainol A, (+)-vaticahainol C, and (−)-albiraminol B. Confirmation of the correlations between the reported spectroscopic data and those obtained by the asymmetric total syntheses provided new insight into the biosynthetic pathway of resveratrol dimers. This study demonstrated that the computation-guided approach can be a powerful strategy to advance natural product chemical research. The concept of computation-guided natural product

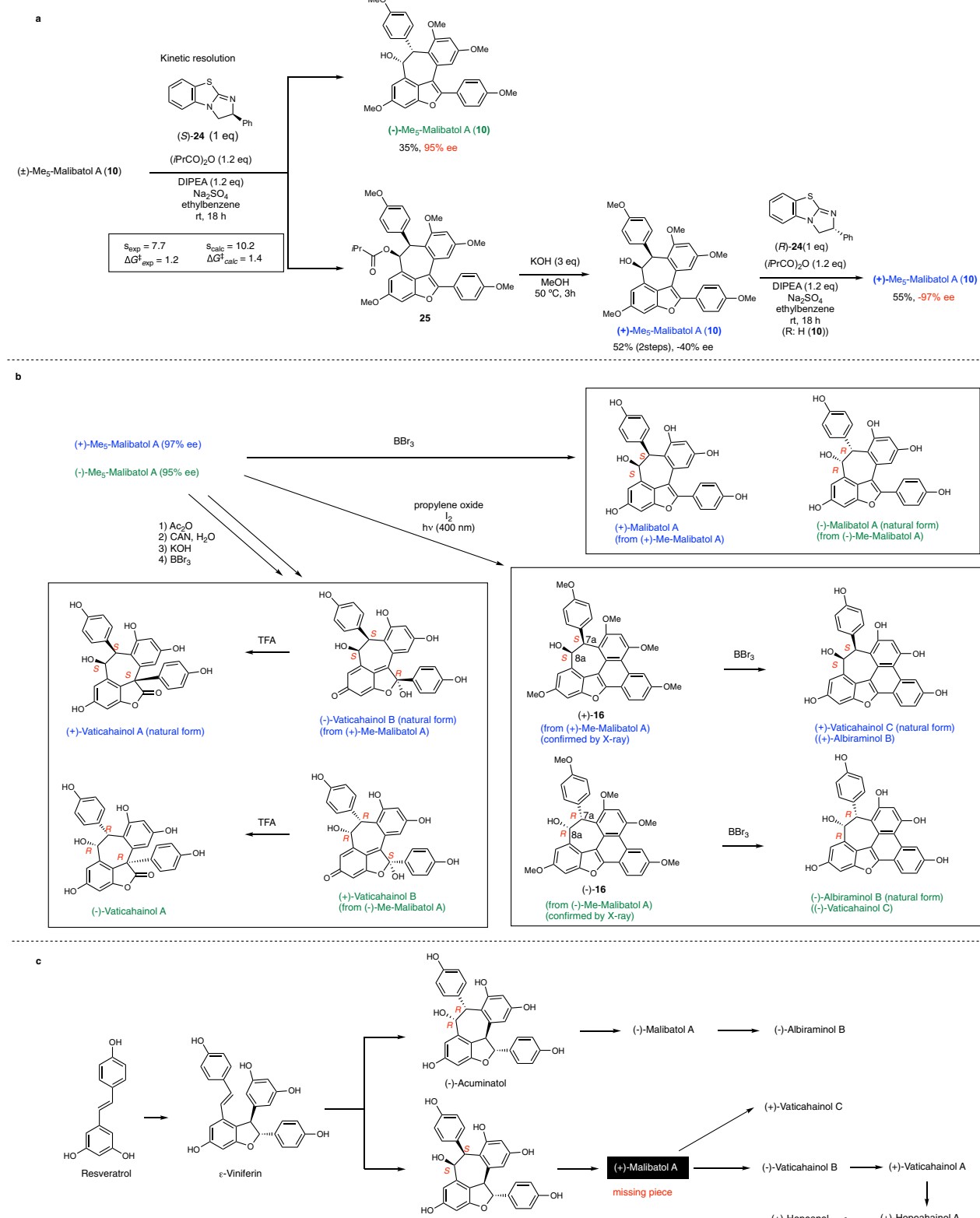

**Fig. 5 Kinetic resolution and asymmetric total syntheses of resveratrol dimers. a** Kinetic resolution of (±)-Me₅-malibatiol A (**10**) to obtain optically active (+)-**10** and (−)-**10**. **b** Asymmetric total syntheses of (+)- and (−)- resveratrol dimers and determination of their absolute configurations. **c** Proposed biosynthetic pathways. DIPEA N,N-diisopropylethylamine, ee enantiomeric excess.

synthesis is expected to be standardized in the future as a powerful research method in the field of organic synthesis.

## Methods

**General information**. NMR spectra were recorded on a JEOL ecs 400, ecz 400, ecz 600, eca 600 spectrometer. Chemical shifts in CDCl$_3$, acetone-d$_6$, DMSO-d$_6$ or CD$_3$OD were reported downfield from tetramethylsilane (TMS) (= 0 ppm) or solvent signal [acetone-d$_6$ (= 2.04 ppm), DMSO-d$_6$ (= 2.49 ppm) or CD$_3$OD (= 3.30 ppm)] for $^1$H NMR. Data are reported as follows: chemical shift, multiplicity (s = singlet, d = doublet, t = triplet, m = multiplet, and br = broad), integration and coupling constants in Hz. For $^{13}$C NMR, chemical shifts were reported in the scale relative to the solvent signal [CHCl$_3$ (77.0 ppm), acetone-d$_6$ (29.8 ppm), DMSO-d$_6$ (39.5 ppm) or CD$_3$OD (49.0 ppm)] as an internal reference. ESI mass spectra were measured on JEOL AccuTOF LC-plus JMS-T100LP. Optical rotations were measured on a JASCO P-1020 polarimeter. The enantiomeric excess (ee) was determined by HPLC analysis. HPLC was performed on JASCO HPLC systems consisting of the following: pump, PU-980; detector, UV-970; column DAICEL CHIRALPAK OD-3; mobile phase, n-hexane/$i$-PrOH. Analytical thin layer chromatography was performed on Kieselgel 60F254, 0.25 mm thickness plates. Column chromatography was performed with silica gel 60 N (spherical, neutral 63-210 mesh). Reactions were conducted in dry solvent. Other reagents were purified by the usual methods.

## Data availability

The Crystallographic data generated in this study have been deposited in the Cambridge Crystallographic Data Centre under accession code CCDC 2080017 (compound epi-6), CCDC 2080023 (compound 13), CCDC 2080016 (compound (−)-16), and CCDC 2080022 (compound (+)-16) [https://www.ccdc.cam.ac.uk/structures/]. All other data generated in this study are provided in the Supplementary Information/Source Data file. Source data are provided with this paper.

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

## Acknowledgements

Numerical calculations were carried out on SR24000 at Institute of Management and Information Technologies, Chiba University, Japan. We thank Dr. Tomoki Yoneda (Hokkaido university) for assistance and advice regarding X-ray crystallographic analysis.

## Author contributions

M.N. conceived, designed, and performed all calculations. M.N. and Y.A. carried out the synthetic experiments. All authors discussed and co-wrote the manuscript. Correspondence and requests for materials should be addressed to M.N and T.N.

## Competing interests

The authors declare no competing interests.
