## [Peer Review File · Nature Communications]

REVIEWER COMMENTS

Reviewer #1 (Remarks to the Author):

I am excited to see computational quantum chemistry being applied to aid synthesis design. This is the most noteworthy aspect of the manuscript and one that I applaud.

Nonetheless, there are issues with the computational work that should be addressed (see below), the synthetic methods applied do not appear to be new, and the conclusions made seem to me to go too far.

A. Many additional references should be added to this paper to provide appropriate context. For example:

1. Newhouse and Tantillo have published reviews in Chem Soc Rev about applying computations to synthesis. These would be more appropriate to cite than the current ref 1.

2. Tantillo and Houk have published multiple reviews on using QM calculations to probe biosynthetic pathways. These would be more appropriate to cite than the current ref 2.

3. References could be added on biomimetic synthesis (top p. 3).

4. References could be added on polyphenol bioactivities (middle p. 3).

5. References could be added on difficulties in assigning RO structures (bottom p. 3).

6. References could be added on the synthetic reactions used, eg. Mizoroki-Heck; most synthetic methods are not accompanied by relevant references.

7. On p.8 the original papers reporting spectra should be cited when matches between synthetic and isolated compounds are mentioned.

8. References could be added on light-induced 6-electron electrocyclizations (top p. 9).

9. References could be added on the utility of natural product derivatization (middle p. 9).

10. References could be added on difficulties of late-stage kinetic resolutions (middle p. 11).

11. References for computational methods should be given: Gaussian 16, wB97XD, LANL2DZ, SMD, IRC.

B. With regard to the computations, no information is given justifying the chosen level of theory. Is that level supported by literature for closely related reactions? Was benchmarking carried out?

C. There does not appear to be information on conformational searching or sampling of configurations for bound solvent molecules. These are important issues for molecules of the size and flexibility treated here.

D. On p.4 the authors state that "all possible pathways..." I do not believe that statement to be true. There certainly are other pathways that could have been examined.

E. The authors seem to be assuming that all the biosynthetic reactions happen in water, in the absence of enzymes. Where is the evidence for that assumption?

F. On p.4 "synthesized in silico" is unusual terminology.

G. How is "strain energy" defined (p.5). This issue has been the source of much debate over the years.

H. The authors refer to potential energy on p.6. Why not use the more relevant free energies? Do the authors expect there to be problems with computing them?

I. The authors repeatedly claim to have proven/confirmed biosynthetic pathways. That is not the

case. They have proposed pathways that are supported by their computational and experimental results, which do not account for enzymes and subtleties of biological surroundings. That does not mean they are wrong, but their conclusions are not justified given the available evidence.

J. Figure 1b is confusing without structures. I do not think it adds value to combine several different proposed reactions in this way.

K. When DFT calculations are first mentioned, the specific methods used should be stated or the methods section should be called out.

Reviewer #2 (Remarks to the Author):

In this manuscript, the authors provided an example where a comprehensive computational chemistry survey of the possible diastereomers involved in biosynthetic pathways reveals potential structure determination problem, which is then proved by total synthesis and XRD of the final product. A revised plausible biosynthetic pathway is proposed based on the asymmetric total synthesis, achieved via kinetic resolution, of the relevant natural products. This approach could be meaningful in future natural product analysis, biosynthetic pathway studies and biomimetic total synthesis studies. However, there are some problems in this manuscript the authors need to address before publishing.

Considering the natural products are isolated in optically active form, it's plausible that enzyme is involved in some stage of the transformation and affected the stereoselectivity. Could the author provide some insights to justify the assumption of non-enzymatic environment and using hydroperoxy radical as substrate (reagent) in the simulation?

The authors need to show that they have (taken effort to) found the optimal structure for the intermediates and transition states (instead of a local minima). For example, intuitively, the coordinate shown for TS1 seems unlikely to be the optimal. And a 20 min calculation, shown below, finds two lower local minima for TS1 at the level used in the article (ω B97X-D/6-31G(d,p) with SMD in water). Considering some of the energy differences considered in the article is not large, this issue needs to be addressed for all relevant INTs and TSs.

In Figure 4, the author claims the experimental result validates the computational prediction. However, none of the computational result were validated by experiment (the closest one uses MeCO⁻ in computation and iPrCO⁻ in experiment). To support this claim, the author should list the experimentally measured e.e. data for the 4 entries along side the computational data for comparison, compute the case (iPrCO⁻) used in the synthesis, and check the differences.

The author calculates the kinetic resolution reaction in gas phase ("The other calculations were performed in gas phase."), while the reaction involves highly charged transition states. This could cause significant error due to solvation free energy and may not be cancelled in $\Delta\Delta G^\ddagger$.

Please use a reasonable number of significant digits for computational result. For example, in Figure 4, the $\Delta\Delta G^\ddagger$ was written to 0.01 kcal/mol, and the ratio of kinetic constants were written to an accuracy of 3‰ (equivalent to 0.001 kcal/mol $\Delta\Delta G^\ddagger$). The computational level used there would give far less accuracy. The AConf benchmark gives an estimated RMSD of 0.3 kcal/mol for the level of ω B97X-D/def2-QZVP. Deviation for data in Figure 4 should be far higher than this number with charged transition states, lower (double- ζ) basis set, and more complicated interactions. The ratio of kinetic constant should have at most one significant digit.

Figure 5a is misleading. Please draw it as two separate reactions and make it clear what's the substrate for the second arrow.

Flack parameter for at least one of the crystal is unsatisfactory for absolute configuration determination (with the 3σ range covers the 0.5 point).

Reviewer #3 (Remarks to the Author):

The authors have described a multidisciplinary project that crosses mechanistic study and computational analysis with organic synthesis to deliver a groundbreaking study in biosynthesis. This work also demonstrates the utility in conducting calculations as a way to establish the most viable means to synthesize a natural product. This article should be accepted after some important revisions noted below.

1. The authors need to include a discussion of the level of theory that was employed. This level of theory should be benchmarked to some set of experiments or a reference should be provided that benchmarks this level of theory. In general, I would have expected single point calculations to be conducted at a higher level of theory than what was performed.

2. I suggest removing Figure 3c as it does not add value and only detracts from otherwise great work. If the authors wanted to make impact here, then they should synthesize ~50 analogs that increase the polarity and evaluate the biological effects of those compounds. Otherwise, it is not surprising, interesting, or useful to show a series of random cross coupling reactions.

3. The calculations in Figure 4 are not validated by the experimental findings. There is a single experimental data point compared to a calculated value. One would need to see a trend across several substrates or several catalysts (ideally with the same mechanism) to know if there is agreement. The calculated difference in energy between the transition states of 0.74 versus the experimentally determined value (1.23) do agree, but this may be by accident. A difference of 0.5 kcal/mol is acceptable, but note that the second best catalyst is 0.29, which is also a 0.5 kcal/mol difference. Obtaining ee values for several different commercially available catalysts for the same transformation of starting material to product would not be challenging. At a minimum, the catalysts in Figure 4 should be attempted and ee values determined. This would at least validate the hypothesis that catalyst A is optimal.

4. ¹³C NMR data looks like it was automatically imported and not checked. This should be thoroughly re-examined.

5. Alert Level B should be addressed for X-ray structure.

6. Comparisons of natural and synthetic NMR data should be tabulated. These must be included before publication anywhere.

We thank the reviewers for helping us to greatly improve our manuscript. All of the authors appreciate the thoughtful and helpful comments. Please find our responses to the reviewers below.

Reviewer #1

A. Many additional references should be added to this paper to provide appropriate context. For example:

1. Newhouse and Tantillo have published reviews in *Chem Soc Rev* about applying computations to synthesis. These would be more appropriate to cite than the current ref 1.

Our response:

We added the following references as suggested.

- 1 Tantillo, D. J. Questions in natural products synthesis research that can (and cannot) be answered using computational chemistry. *Chem. Soc. Rev.* **47**, 7845-7850, doi:10.1039/c8cs00298c (2018).
- 2 Elkin, M. & Newhouse, T. R. Computational chemistry strategies in natural product synthesis. *Chem. Soc. Rev.* **47**, 7830-7844, doi:10.1039/c8cs00351c (2018).
- 3 Bemis, C. Y. *et al.* Total Synthesis and Computational Investigations of Sesquiterpene-Tropolones Ameliorate Stereochemical Inconsistencies and Resolve an Ambiguous Biosynthetic Relationship. *J. Am. Chem. Soc.* **143**, 6006-6017, doi:10.1021/jacs.1c02150 (2021).

2. Tantillo and Houk have published multiple reviews on using QM calculations to probe biosynthetic pathways. These would be more appropriate to cite than the current ref 2.

Our response:

We added the following reference as suggested.

- 4 Tantillo, D. J. Biosynthesis via carbocations: Theoretical studies on terpene formation. *Nat. Prod. Rep.* **28**, 1035-1053, doi:10.1039/c1np00006c (2011).

3. References could be added on biomimetic synthesis (top p. 3).

Our response:

We added the following reference as suggested.

- 6 Elkin, M., Scruse, A. C., Turlik, A. & Newhouse, T. R. Computational and Synthetic Investigation of Cationic Rearrangement in the Putative Biosynthesis of Justicane Triterpenoids. *Angew. Chem., Int. Ed.* **58**, 1025-1029, doi:10.1002/anie.201810566 (2019).

4. References could be added on polyphenol bioactivities (middle p. 3).

Our response:

We added the following references as suggested.

- 9 Qiao, H. *et al.* Antitumor effects of naturally occurring oligomeric resveratrol derivatives. *Faseb J.* **27**, 4561-4571, doi:10.1096/fj.13-231613 (2013).
- 10 Shen, J. *et al.* Update on phytochemistry and pharmacology of naturally occurring resveratrol oligomers. *Molecules* **22**, 2050/2051-2050/2026, doi:10.3390/molecules22122050 (2017).

5. References could be added on difficulties in assigning RO structures (bottom p. 3).

Our response:

In general, structural determination of complex natural products is difficult and considerable effort is required. From this point of view, the content of the original description is correct. We were unable to find any specific literature describing the difficulties in determining the structures of resveratrol oligomers. Therefore, we changed the sentence as follows:

“Owing to their structural complexity deriving from the diverse oligomerization patterns and large number of stereocenters, as well as the low amounts isolable from plant resources, considerable effort is required for their structural determination by natural product chemistry research techniques.

6. References could be added on the synthetic reactions used, eg. Mizoroki-Heck; most synthetic methods are not accompanied by relevant references.

Our response:

We deleted Fig.3c, which included synthetic reactions, e.g., Mizoroki-Heck, as Reviewer #3 suggested.

7. On p.8 the original papers reporting spectra should be cited when matches between synthetic and isolated compounds are mentioned.

Our response:

We added the citation as suggested.

8. References could be added on light-induced 6-electron electrocyclizations (top p. 9).

Our response:

We added the following reference as suggested.

40 Chan, C.-K., Chen, Y.-C., Chen, Y.-L. & Chang, M.-Y. Synthesis of substituted phenanthrofurans. *Tetrahedron* **71**, 9187-9195, doi:10.1016/j.tet.2015.10.060 (2015).

9. References could be added on the utility of natural product derivatization (middle p. 9).

Our response:

We deleted Fig.3c, which included synthetic reactions, e.g., Mizoroki-Heck, as Reviewer #3 suggested. Therefore, sentences explaining the utility of natural product derivatization were also deleted.

10. References could be added on difficulties of late-stage kinetic resolutions (middle p. 11).

Our response:

In general, late-stage functionalization of complex molecules is a challenging task in organic synthesis. From this point of view, the content of the sentence on the difficulties of late-stage kinetic resolution would be correct. Because we were unable to find any specific literature on this topic, however, we deleted the sentence.

11. References for computational methods should be given: Gaussian 16, ωB97X-D, LANL2DZ, SMD, IRC.

Our response:

We added the following references, as suggested.

25 Frisch, M. J. *et al.* Gaussian 16, Revision C.01. Gaussian, Inc., Wallingford CT (2016)

- 26 Chai, J.-D. & Head-Gordon, M. Long-range corrected hybrid density functionals with damped atom-atom dispersion corrections. *Phys. Chem. Chem. Phys.* **10**, 6615-6620, doi:10.1039/b810189b (2008).
- 27 Hay, P. J. & Wadt, W. R. Ab initio effective core potentials for molecular calculations. Potentials for the transition metal atoms scandium to mercury. *J. Chem. Phys.* **82**, 270-283, doi:10.1063/1.448799 (1985).
- 28 Wadt, W. R. & Hay, P. J. Ab initio effective core potentials for molecular calculations. Potentials for main group elements sodium to bismuth. *J. Chem. Phys.* **82**, 284-298, doi:10.1063/1.448800 (1985).
- 29 Hay, P. J. & Wadt, W. R. Ab initio effective core potentials for molecular calculations. Potentials for potassium to gold including the outermost core orbitals. *J. Chem. Phys.* **82**, 299-310, doi:10.1063/1.448975 (1985).
- 30 Marenich, A. V., Cramer, C. J. & Truhlar, D. G. Universal Solvation Model Based on Solute Electron Density and on a Continuum Model of the Solvent Defined by the Bulk Dielectric Constant and Atomic Surface Tensions. *J. Phys. Chem. B* **113**, 6378-6396, doi:10.1021/jp810292n (2009).
- 31 Fukui, K. The path of chemical reactions - the IRC approach. *Acc. Chem. Res.* **14**, 363-368, doi:10.1021/ar00072a001 (1981).
- 32 Shao, Y. *et al.* Advances in methods and algorithms in a modern quantum chemistry program package. *Phys. Chem. Chem. Phys.* **8**, 3172-3191, doi:10.1039/b517914a (2006).

B. With regard to the computations, no information is given justifying the chosen level of theory. Is that level supported by literature for closely related reactions? Was benchmarking carried out? Our response:

We benchmarked the functionals/basis sets using the structures of malibatol A, (epi-)vaticahainol A and B. We started the benchmarking study by searching their configurations using Spartan 18 (MMFF, Monte Carlo, within 25 kcal/mol, up to 1000 conformers). All the calculated conformations with the MMFF level calculated (73, 67, 95, 29, and 35 conformers for malibatol A, vaticahainol A, epi-vaticahainol A, vaticahainol B, and epi-vaticahainol B, respectively) were subjected to structural optimization, and vibrational frequency calculation at the ω B97X-D/6-31G** (SMD, water) DFT level using the Gaussian16 program, and the global minimum was determined.

Global minimum of Malibatol A

Global minimum of epi-Vaticahainol B

Global minimum of Vaticahainol B

Global minimum of Vaticahainol A

Global minimum of epi-Vaticahainol A

Then, using the global minimum of malibatol A, we performed structural optimization studies with various functions/6-31G** (SMD, water). We also optimized the structure with ab initio calculation: MP2/6-311G** (SMD, water). We then compared the optimized geometry of MP2/6-311G** and various functions/6-31G** by the root-mean-square deviation (RMSD).

Method	ab initio (MP2)	DFT												
		B3LYP	B3PW91	B98	BLYP	cam-B3LYP	mPW1PW91	PBEPBE	PBE1PBE	M06	M06-L	M06-2X	wB97	wB97X-D
Functional	6-311G**	6-31G**												
Basis set	0.00	0.37	0.35	0.34	0.40	0.34	0.32	0.34	0.31	0.29	0.20	0.19	0.26	0.17
RMSD (Å)														

Method	ab initio (MP2)	DFT (wB97X-D)										
		STO-3G	6-31G*	6-31G**	6-31+g**	6-311g**	6-311+g**	cc-pvDZ	aug-cc-pvDZ	cc-pvTZ	def2SVP	def2TZVP
Basis set	6-311G**											
RMSD (Å)	0.00	0.21	0.17	0.17	0.16	0.15	0.14	0.19	0.14	0.20	0.17	0.19

As a result, the value of ω B97X-D (RMSD = 0.17 Å) was closest to the ab initio MP2 calculation. Thus, we selected ω B97X-D as the functional. Next, we compared the optimized geometry of MP2/6-311G** (SMD, water) and various ω B97X-D/various basis sets (SMD, water) by RMSD. As a result, 6-311+G** and aug-ccpvdz showed the smallest RMSD (0.14Å). On the other hand, the RMSD of 6-31G** was 0.17 Å, which is only 0.03 Å different from that of 6-311+G** and aug-ccpvdz. Therefore, we decided to use 6-31G** as a basis set in view of the computational costs. Based on the above benchmarking, we adopted ω B97X-D/6-31G** (SMD, water) for the structural optimization.

Opt (SMD in water)	Functional	wB97X-D							
	Basis set	6-31G**							
SP (SMD in water)	Method	ab initio (MP2)				–	DFT (wB97X-D)		
	Basis set	6-311+G**	cc-pvTZ	def2TZVP	average	–	6-311+G**	cc-pvTZ	def2TZVP
ΔE (Vaticahainol A - epi-Vaticahainol A) (kcal/mol)		-2.1	-2.4	-2.6	-2.4	-2.3	-3.1	-3.3	-3.4
ΔE (Vaticahainol B - epi-Vaticahainol B) (kcal/mol)		2.3	2.3	2.2	2.3	2.7	2.0	1.9	1.7
Mean absolute error (kcal/mol)					0.0	0.3	0.5	0.7	0.8

Global minimum of Vaticahainol A

Global minimum of epi-Vaticahainol A

Global minimum of epi-Vaticahainol B

Global minimum of Vaticahainol B

Next, we performed benchmarking studies of energy corrections by single-point calculations. First, we performed ab initio single-point calculations at the ab initio MP2/6-311+G**, cc-pvTZ, and def2TZVP levels (SMD, water) using the global minimum structure of (epi-)vaticahainol A and B optimized at the ω B97X-D /6-31G** (SMD, water) level. The energy difference (ΔE) between vaticahainol A or B and their epimer was compared. Vaticahainol A and its epimer showed a 2.1-2.6 kcal/mol energy difference (the energy of vaticahainol A was 2.4 kcal/mol [average] lower), and vaticahainol B and its epimer showed a 2.2-2.3 kcal/mol energy difference (the energy of vaticahainol B was 2.3 kcal/mol [average] higher). We then performed single-point energy calculations at the ω B97X-D/6-311+G**, cc-pvTZ, and def2TZVP levels (SMD, water) and compared the results with the average of the ab initio calculations. The mean absolute error (MAE) of the difference from the energy correction by MP2 was within 0.8 kcal/mol for all basis sets. Because the lowest MAE was the original data without energy corrections, however, we decided not to correct the energy by a single-point calculation.

We added the following sentences:

“The conformation search was performed by the Monte Carlo method with the Spartan18 program using an energy within 25 kcal/mol and an upper limit of 1000 conformers at the MMFF level. All the calculated configurations were subjected to structural optimization and vibrational frequency calculation at the DFT level using the Gaussian16 program, and the global minimum was determined.”

“Before calculating the biosynthetic pathway, the global minimum (SM) of malibatol A was calculated by a conformational search (see Supplementary Fig. X).”

Also, the results of the benchmarking study were added to the SI.

C. There does not appear to be information on conformational searching or sampling of

configurations for bound solvent molecules. These are important issues for molecules of the size and flexibility treated here.

Our response:

We performed the conformation search as described above, and all transition states in Fig. 2 were re-calculated using global minimum structures of malibatol A or (epi-)vaticahainol A, or B. All biosynthetic calculations were performed in water (SMD), but no solvent molecules were added, except for in the protonation reaction. In TS3 and TS3' shown in Fig. 2a, proton shuttling did not occur with the addition of one or two water molecules. The hydrogen-bonding network formed when more than three water molecules were added, and the activation barrier was low enough to occur in nature. Of course, the substrates are surrounded by many water molecules in an actual biological reaction, but we concluded that the addition of three water molecules is enough to explain the proton shuttling in Fig. 2a.

D. On p.4 the authors state that "all possible pathways..." I do not believe that statement to be true. There certainly are other pathways that could have been examined.

Our response:

We deleted the words and changed the sentence as follows:

“All possible ~~synthetic pathways~~ and natural and non-natural diastereomers were comprehensively simulated...”

E. The authors seem to be assuming that all the biosynthetic reactions happen in water, in the absence of enzymes. Where is the evidence for that assumption?

Our response:

This is a significant point. To our knowledge, there are no reports of the enzymes involved in the biosynthesis of vaticahainols, but cytochrome P450s are reportedly involved in oxidation reactions in many plants and animals and catalyze the in vivo epoxidation of olefins. Therefore, we calculated the oxidation reaction by the O=Fe(porphyrin)(SMe) complex (compound I: cpd I) as a model structure of P450, considering enzymatic epoxidation (Below figure). The numbers in parentheses are the energy of the quartet-state.

We attempted to calculate the transition state for epoxidation by cpd I based on the calculated global minimum structure (SM). For the biosynthesis of vaticahainol B, the epoxidation must proceed from the same side as the Ar group on C7a. Due to the steric hindrance of the pseudo-axial Ar group on C7a in SM, the global minimum of malibatol A, the bulky cpd I could not be approached. Therefore, no transition state for this approach mode to SM was found. Instead, the desired epoxidation transition state (TS1) was found from the local minimum SM2, where the seven-membered ring of the global minimum flips and the Ar group of C7a is in the pseudo-equatorial position (blue energy diagram). On the other hand, a transition state in which epoxidation proceeds from the opposite side of the pseudo-axial Ar group on C7a was found from the global minimum (SM). This transition state yields the C7b epimer of vaticahainol B, which has not been isolated as a natural product. Because of the conformational difference, the activation barrier of TS1 was 8.7 kcal/mol higher than that of TS1', indicating that the biosynthesis of C7b-epi-vaticahainol B was kinetically favored. The reactions with the quartet state of iron were also calculated and showed a similar trend (energies in parentheses). This result is considered to be a more plausible representation of the biosynthetic pathway. Thus, we moved the non-enzymatic OOH radical pathway to the SI, and the enzymatic result is shown in Fig 2a.

We also added the following references.

- 33 Shaik, S. *et al.* P450 enzymes: their structure, reactivity, and selectivity-modeled by QM/MM calculations. *Chem Rev* **110**, 949-1017 (2010).

- 34 Shaik, S., Milko, P., Schyman, P., Usharani, D. & Chen, H. Trends in aromatic oxidation reactions catalyzed by cytochrome P450 enzymes: a valence bond modeling. *J. Chem. Theory Comput.* **7**, 327-339, doi:10.1021/ct100554g (2011).
- 35 Grandner, J. M., Cacho, R. A., Tang, Y. & Houk, K. N. Mechanism of the P450-Catalyzed Oxidative Cyclization in the Biosynthesis of Griseofulvin. *ACS Catal.* **6**, 4506-4511, doi:10.1021/acscatal.6b01068 (2016).

F. On p.4 "synthesized in silico" is unusual terminology.

Our response:

We changed the words as follows:

“calculated ~~synthesized in silico~~ starting from vaticahainol A”

G. How is "strain energy" defined (p.5). This issue has been the source of much debate over the years.

Our response:

The figure and discussion of strain energies were deleted because the relevant part of Fig. 2a was a calculation with a model molecule of P450 and moved to the SI.

H. The authors refer to potential energy on p.6. Why not use the more relevant free energies? Do the author expect there to be problems with computing them?

Our response:

We corrected the sentence as follows:

“The free ~~potential~~ energy of vaticahainol A...”

I. The authors repeatedly claim to have proven/confirmed biosynthetic pathways. That is not the case. They have proposed pathways that are supported by their computational and experimental results, which do not account for enzymes and subtleties of biological surroundings. That does not mean they are wrong, but their conclusions are not justified given the available evidence.

Our response:

We changed the sentences as follows:

“Therefore, it was synthetically and computationally suggested ~~confirmed~~ that the correctly identified vaticahainol A...”

“the C7a, C8a epimer of (–)-acuminatol, is ~~proposed~~ suggested to be the biosynthetic intermediate...”

“natural (+)-vaticahainol A is the plausible biosynthetic intermediate”

J. Figure 1b is confusing without structures. I do not think it adds value to combine several different proposed reactions in this way.

Our response:

We deleted Fig. 1a and added the structure to Fig. 1b.

K. When DFT calculations are first mentioned, the specific methods used should be stated or the methods section should be called out.

Our response:

We added descriptions of the computational methods in the Methods, as indicated.

Reviewer #2

A. Considering the natural products are isolated in optically active form, it's plausible that enzyme is involved in some stage of the transformation and affected the stereoselectivity. Could the author provide some insights to justify the assumption of non-enzymatic environment and using hydroperoxy radical as substrate (reagent) in the simulation?

Our response:

Please see our response to the comment E of Reviewer 1.

B. The authors need to show that they have (taken effort to) found the optimal structure for the intermediates and transition states (instead of a local minima). For example, intuitively, the coordinate shown for TS1 seems unlikely to be the optimal. And a 20 min calculation, shown below, finds two lower local minima for TS1 at the level used in the article (ω B97X-D/6-31G(d,p) with SMD in water). Considering some of the energy differences considered in the article is not large, this issue needs to be addressed for all relevant INTs and TSs.

Our response:

As described in our response to comment B of Reviewer 1, we calculated the global minimums for malibatol A, (epi)-vaticahainol A and B by conformational searching.

Based on the calculated global minima, all the transition states in Fig. 2 were re-calculated. As a result, the overall energy diagram is now lower than in the previous version.

TS1 in Fig.2a was changed to a transition state with a model molecule of P450 as described above, and the reaction with the non-enzymatic OOH radical was moved to the SI. The conformational search for the transition state of epoxidation by non-enzymatic OOH radicals pointed out in this comment, however, was investigated using the global minimum of malibatol A.

num	Opt/Freq		TS1 (ΔG)						TS1' (ΔG)				$\Delta\Delta G^\ddagger$ ($\Delta G_{TS1} - \Delta G_{TS1'}$)
	Functional	Basis set	A	B	C	D	E	F	G	H	I	J	
1	wB97X-D	6-31G**	0.6	2.6	5.4	3.4	3.6	3.8	2.3	1.4	0.0	0.2	0.9
2	wB97		0.4	2.3	5.6	3.8	3.9	3.7	2.5	1.7	0.0	0.5	0.7
3	B3LYP		2.6	3.3	5.6	4.4	3.9	3.6	1.9	1.1	0.1	0.0	2.8
4	B98		3.2	4.1	6.2	5.4	5.0	5.1	2.9	2.2	1.3	0.0	3.1
5	BLYP		3.0	4.2	6.0	5.1	4.2	4.3	2.2	1.3	0.6	0.0	3.1
6	B3PW91		2.7	3.7	6.2	4.8	4.4	4.1	2.2	1.6	0.7	0.0	2.7
7	cam-B3LYP		2.1	2.9	5.3	4.1	3.9	4.0	2.0	1.4	0.7	0.0	2.1
8	mPW1PW91		1.7	2.4	5.3	4.3	3.9	3.6	1.7	1.0	0.0	0.3	1.9
9	M06		1.0	2.0	5.6	4.2	3.6	3.5	1.4	1.0	0.0	0.5	1.2
10	M06-L		1.5	2.6	6.3	5.6	3.7	3.4	2.3	1.4	0.3	0.0	1.7
11	M06-2X		0.0	2.4	4.3	3.6	4.3	4.2	3.1	2.4	0.4	1.0	-0.2
12	PBEPBE		1.1	3.2	5.0	3.7	3.4	3.5	1.7	1.2	0.4	0.0	1.4
13	PBE1PBE		1.9	3.5	5.9	4.8	4.6	4.5	2.5	1.6	0.1	0.0	2.3
14	HF		4.2	4.9	6.5	5.4	3.7	3.8	6.7	2.9	0.0	0.1	3.5
15	wB97X-D	def2SVP	0.2	2.4	5.2	3.8	4.1	3.9	2.5	1.6	0.4	0.0	0.5
16	wB97X-D	cc-pvdZ	0.8	2.8	5.9	4.0	3.6	3.9	2.9	1.2	0.3	0.0	1.1

As the result of changing the dihedral of CC-OO and CO-OH in transition states, we found six TS1s (A-F) and four TS1' (G-J). By considering the Boltzmann distribution and calculating ΔG^\ddagger , we revealed that TS1', which affords epi-vaticahainol B, gives a 0.9 kcal/mol smaller activation energy (run 1). To check that this value ($\Delta\Delta G^\ddagger = 0.9$ kcal/mol) is not specific to the ω B97X-D/6-31G(d,p) level calculation, we calculated all 10 transition states using various functions/6-31G(d,p) (run 2-14) and ω B97X-D/def2SVP or cc-pvdz (run 15, 16). The results

showed that TS1' gave smaller activation energies than TS1 in all calculations except M06-2X, indicating that the generation of epi-vaticahainol B would be kinetically favored. We added these results to the SI.

C. In Figure 4, the author claims the experimental result validates the computational prediction. However, none of the computational result were validated by experiment (the closest one uses MeCO- in computation and iPrCO- in experiment). To support this claim, the author should list the experimentally measured e.e. data for the 4 entries along side the computational data for comparison, compute the case (iPrCO-) used in the synthesis, and check the differences.

Our response:

An experimental KR study was attempted to confirm whether the calculated tendency correlates with the experimental results. When we used acetic anhydride (R = Me), the KR was not successful with any chiral reagent due to background reactions. The use of isobutyric anhydride (R = iPr) prevented the background reaction to yield the s value: chiral acylating complex A showed the best $\Delta\Delta G_{TS}$ and s value (1.3 kcal/mol and 8.8, respectively), and B-D showed a small $\Delta\Delta G_{TS}$ and s value (-0.1 to -0.3 kcal/mol and 1.1 to 1.6, respectively), suggesting that A would be the best reagent for this KR as predicted by model calculation. We added the information to Fig. 4.

Chiral reagent	Absolute configuration	TS	ΔG^\ddagger								
			1	2	3	4	5	6	7	8	9
	(S, S)	iPr_SS_A1	5.2	2.0	6.0	4.6	1.9	6.0	5.3	2.3	5.7
		iPr_SS_A2	0.5	5.0	7.7	1.0	5.8	8.1	1.2	7.5	9.6
		iPr_SS_A3	5.3	12.4	5.9	6.3	9.3	4.0	6.5	10.9	5.7
		iPr_SS_A4	6.3	9.3	4.1	8.8	6.4	5.7	10.9	5.3	2.8
		iPr_SS_A5	0.9	12.0	10.9	1.9	12.7	10.6	2.2	13.4	10.7
		iPr_SS_A6	12.4	15.5	8.3	11.5	15.5	0.9	13.4	1.9	8.8
		iPr_SS_A7	1.2	6.6	6.0	0.0	6.6	6.3	2.2	6.2	6.4
	(R, R)	iPr_RR_A1	8.4	12.8	6.2	9.9	11.3	5.8	8.7	7.2	5.8
		iPr_RR_A2	1.5	10.5	10.6	1.9	8.6	9.8	1.5	8.5	10.3
		iPr_RR_A3	4.3	8.8	6.6	4.2	7.9	6.8	3.2	8.9	7.1
		iPr_RR_A4	2.8	7.9	7.3	2.1	8.8	8.4	3.3	7.6	7.8
		iPr_RR_A5	8.9	9.4	16.7	9.0	16.8	7.1	16.9		
		iPr_RR_A6	4.1	4.3	6.6	3.8	7.1	4.3	5.9	7.2	3.8

Also, we performed the calculation again using complex A (R = iPr). Due to the rotational isomers of the iPr group, we considered 115 conformers (63 TSs for (S,S) and 52 TSs for (R,R))

and obtained $s = 10.2$ and $\Delta G^\ddagger = 1.4$, which showed a correlation with the experimental results ($s = 8.1$ and $\Delta G^\ddagger = 1.2$ kcal/mol).

We added the information to Figs. 5.

D. The author calculates the kinetic resolution reaction in gas phase (“The other calculations were performed in gas phase.”), while the reaction involves highly charged transition states. This could cause significant error due to solvation free energy and may not be cancelled in $\Delta\Delta G^\ddagger$.

Our response:

This comment is significant. If a computational chemistry validation is conducted based on the actual experimental data, we think it is reasonable to use the specific solvent for the calculation, as this reviewer suggested. We performed the calculations in the gas phase, however, because the calculations were performed prior to the experiment, i.e., the stage at which we had no information on the solvent effect. We expected that the $\Delta\Delta G$ derived from steric hindrance of the asymmetric catalyst could be estimated even in the gas phase, and hoped that this would be a guideline for choosing an asymmetric catalyst. As described in our previous response, our predictions correlated well with the experimental results. Thus, we concluded that the gas-phase calculations can be used to predict the results.

E. Please use a reasonable number of significant digits for computational result. For example, in Figure 4, the $\Delta\Delta G^\ddagger$ was written to 0.01 kcal/mol, and the ratio of kinetic constants were written to an accuracy of 3% (equivalent to 0.001 kcal/mol $\Delta\Delta G^\ddagger$). The computational level used there would give far less accuracy. The AConf benchmark gives an estimated RMSD of 0.3 kcal/mol for the level of ω B97X-D/def2-QZVP. Deviation for data in Figure 4 should be far higher than this number with charged transition states, lower (double- ζ) basis set, and more complicated interactions. The ratio of kinetic constant should have at most one significant digit.

Our response:

We corrected the digits as suggested.

F. Figure 5a is misleading. Please draw it as two separate reactions and make it clear what's the substrate for the second arrow.

Our response:

We revised Fig 5a and c.

G. Flack parameter for at least one of the crystal is unsatisfactory for absolute configuration determination (with the 3σ range covers the 0.5 point).

Our response:

By solving the X-ray structure of Xray_minus_16 again, we obtained sufficient accuracy to determine the absolute configuration. (refine_ls_abs_structure_Flack -0.02(9))

Reviewer #3

1. The authors need to include a discussion of the level of theory that was employed. This level of theory should be benchmarked to some set of experiments or a reference should be provided that benchmarks this level of theory. In general, I would have expected single point calculations to be conducted at a higher level of theory than what was performed.

Our response:

Please see our response to comment B of Reviewer 1.

2. I suggest removing Figure 3c as it does not add value and only detracts from otherwise great work. If the authors wanted to make impact here, then they should synthesize ~50 analogs that increase the polarity and evaluate the biological effects of those compounds. Otherwise, it is not surprising, interesting, or useful to show a series of random cross coupling reactions.

Our response:

We deleted Fig. 3c as suggested.

3. The calculations in Figure 4 are not validated by the experimental findings. There is a single experimental data point compared to a calculated value. One would need to see a trend across several substrates or several catalysts (ideally with the same mechanism) to know if there is agreement. The calculated difference in energy between the transition states of 0.74 versus the experimentally determined value (1.23) do agree, but this may be by accident. A difference of 0.5 kcal/mol is acceptable, but note that the second best catalyst is 0.29, which is also a 0.5 kcal/mol difference. Obtaining ee values for several different commercially available catalysts for the same transformation of starting material to product would not be challenging. At a minimum, the catalysts in Figure 4 should be attempted and ee values determined. This would at least validate the hypothesis that catalyst A is optimal.

Our response:

Please see our response to comment C of Reviewer 2.

4. ¹³C NMR data looks like it was automatically imported and not checked. This should be thoroughly re-examined.

Our response:

All ¹H and ¹³C NMR were entered manually, and have been rechecked. We added “two carbons are missing at the 25 °C measurement” to a sentence for the ¹³C NMR data of Me₅-hopeahainol A. We measured NMR at -30 °C with hopeahainol A and confirmed that the data were same as in the isolation report and Nicolaou’s synthesis report.

5. Alert Level B should be addressed for X-ray structure.

Our response:

We had added a comment regarding Alert Level B using VRF. Please check the pdf again.

6. Comparisons of natural and synthetic NMR data should be tabulated. These must be included before publication anywhere.

Our response:

We created tables as suggested and added the following tables to the SI.

1H NMR			
H	Synthetic Malibatol A	Natural Malibatol A	Δx (ppm)
1	5.28 (s, 1H)	5.28 (ddd, J = 2.5, 1.0, 1.0 Hz, 1H)	0.00
2	5.47 (s, 1H)	5.46 (dd, J = 2.5, 1.0 Hz, 1H)	0.01
3	6.31 (d, J = 2.8 Hz, 1H)	6.30 (d, J = 2.5 Hz, 1H)	0.01
4	6.33 (d, J = 8.7 Hz, 2H)	6.33 (dd, J = 9.0, 2.5 Hz, 2H)	0.00
5	6.51 (d, J = 2.3 Hz, 1H)	6.51 (d, J = 2.5 Hz, 1H)	0.00
6	6.58 (d, J = 2.3 Hz, 1H)	6.57 (dd, J = 2.0, 1.0 Hz, 1H)	0.01
7	6.81 (d, J = 8.7 Hz, 2H)	6.80 (dd, J = 8.5, 2.5 Hz, 2H)	0.01
8	7.01 – 7.03 (m, 3H)	7.01 (dd, J = 2.0, 1.0 Hz, 1H)	0.01
9		7.02 (dd, J = 9.0, 2.5 Hz, 2H)	0.00
10	7.44 (d, J = 8.7 Hz, 2H)	7.45 (dd, J = 8.5, 2.5 Hz, 2H)	-0.01

13C NMR			
C	Synthetic Malibatol A	Natural Malibatol A	Δx (ppm)
1	49.9	48.8	1.1
2	74.8	74.8	0.0
3	95.9	95.9	0.0
4	102.2	102.2	0.0
5	109.7	109.7	0.0
6	109.9	109.9	0.0
7	114.7	114.7	0.0
8	116.4	116.4	0.0
9	117.3	117.3	0.0
10	119.1	119.1	0.0
11	121.2	121.3	-0.1
12	124.7	124.7	0.0
13	130.6	130.6	0.0
14	130.9	130.9	0.0
15	133.4	133.4	0.0
16	135.8	135.8	0.0
17	139.7	139.7	0.0
18	151.2	151.2	0.0
19	155.2	155.2	0.0
20	155.4	155.4	0.0
21	156.2	156.2	0.0
22	156.7	156.7	0.0
23	157.5	157.5	0.0
24	159.1	159.1	0.0

1H NMR			
H	Synthetic epi-Vaticahainol B	Natural Vaticahainol B	Δx (ppm)
1	4.82 (s, 1H)	4.81 (s, 1H)	0.01
2	5.12 (s, 1H)	5.10 (s, 1H)	0.02
3	5.4 (d, J = 1.4 Hz, 1H)	5.39 (d, J = 1.5 Hz, 1H)	0.01
4	6.18 (d, J = 2.7 Hz, 1H)	6.15 (d, J = 1.5 Hz, 1H)	0.03
5	6.27 (dd, J = 2.0, 2.0 Hz, 1H)	6.26 (s, 1H)	0.01
6	6.45 (d, J = 8.7 Hz, 2H)	6.43 (d, J = 8.5 Hz, 2H)	0.02
7	6.46 (d, J = 2.3 Hz, 1H)	6.44 (d, J = 2.0 Hz, 1H)	0.02
8	6.64 (d, J = 8.7 Hz, 2H)	6.63 (d, J = 8.5 Hz, 2H)	0.01
9	6.66 (d, J = 8.7 Hz, 2H)	6.65 (d, J = 8.5 Hz, 2H)	0.01
10	6.95 (d, J = 8.7 Hz, 2H)	6.95 (d, J = 8.5 Hz, 2H)	0.00
11	7.25 (d, J = 2.3 Hz, 1H)	7.23 (d, J = 2.0 Hz, 1H)	0.02
12	8.69 (s, 1H)	8.67 (s, 1H)	0.02
13	9.06 (s, 1H)	9.04 (br, 1H)	0.02
14	9.44 (s, 1H)	9.43 (br, 1H)	0.01
15	9.61 (s, 1H)	9.59 (br, 1H)	0.02
16	9.68 (s, 1H)	9.67 (s, 1H)	0.01

13C NMR			
C	Synthetic epi-Vaticahainol B	Natural Vaticahainol B	Δx (ppm)
1	44.1	45.9	-1.8
2	71.8	73.5	-1.7
3	97.7	99.4	-1.7
4	105.8	107.6	-1.8
5	109.6	111.3	-1.8
6	114.1	115.8	-1.7
7	114.5	116.1	-1.6
8	114.6	116.3	-1.7
9	119.9	121.7	-1.8
10	122.4	124.1	-1.7
11	124.4	126.1	-1.7
12	127.4	129.0	-1.7
13	128.7	130.4	-1.7
14	128.8	130.5	-1.7
15	129.1	130.7	-1.6
16	130.9	132.6	-1.7
17	144.9	146.6	-1.7
18	151.2	152.9	-1.7
19	155.4	157.0	-1.7
20	155.4	157.0	-1.7
21	155.9	157.6	-1.7
22	157.7	159.4	-1.7
23	169.3	171.0	-1.7
24	187.0	188.6	-1.6

1H NMR			
H	Synthetic epi-Vaticahainol A	Natural Vaticahainol A	Δx (ppm)
1	4.66 (dd, J = 4.8, 4.8 Hz, 1H)	4.67 (t, J = 4.0 Hz, 1H)	-0.01
2	5.21 (d, J = 4.8 Hz, 1H)	5.22 (d, J = 4.0 Hz, 1H)	-0.01
3	5.50 (d, J = 4.8 Hz, 1H)	5.55 (d, J = 4.0 Hz, 1H)	-0.05
4	6.05 (d, J = 9.0 Hz, 2H)	6.05 (d, J = 8.2 Hz, 2H)	0.00
5	6.14 (br, 4H)	6.15 (br, 4H)	-0.01
6	6.42 (d, J = 2.0 Hz, 1H)	6.44 (d, J = 2.0 Hz, 1H)	-0.02
7	6.51 (d, J = 8.3 Hz, 2H)	6.52 (d, J = 8.2 Hz, 2H)	-0.01
8	6.55 (d, J = 2.1 Hz, 1H)	6.56 (d, J = 1.5 Hz, 1H)	-0.01
9	7.06 (d, J = 1.4 Hz, 1H)	7.08 (d, J = 1.5 Hz, 1H)	-0.02
10	7.24 (d, J = 2.1 Hz, 1H)	7.25 (d, J = 2.0 Hz, 1H)	-0.01
11	8.58 (br, 1H)	8.60 (br, 1H)	-0.02
12	9.09 (br, 1H)	9.11 (br, 1H)	-0.02
13	9.39 (br, 1H)	9.42 (br, 1H)	-0.03
14	9.61 (br, 1H)	9.64 (br, 1H)	-0.03
15	10.03 (br, 1H)	10.05 (br, 1H)	-0.02

13C NMR			
C	Synthetic epi-Vaticahainol A	Natural Vaticahainol A	Δx (ppm)
1	44.1	44.0	0.0
2	57.9	57.9	0.0
3	71.1	71.1	0.0
4	96.0	96.0	0.0
5	101.9	101.9	0.0
6	105.8	105.8	0.0
7	109.8	109.7	0.1
8	113.5	113.5	0.0
9	114.4	114.4	0.0
10	114.6	114.6	0.0
11	118.2	118.2	0.0
12	127.1	127.1	0.0
13	129.1	129.1	0.0
14	130.3	130.3	0.0
15	131.1	131.1	0.0
16	137.8	137.8	0.0
17	143.1	143.1	0.0
18	152.7	152.7	0.0
19	153.9	153.9	0.0
20	155.8	155.8	0.0
21	156.0	155.9	0.0
22	157.5	157.4	0.1
23	158.5	158.4	0.1
24	175.7	175.7	0.0

1H NMR					
H	Synthetic Vaticahainol C (Albmiranol B)	Natural Vaticahainol C	Δx (ppm)	Natural Albiraminol B	Δx (ppm)
1	5.03 (br, 1H)	none	none	5.08 (br, 1H)	-0.05
2	5.5 (s, 1H)	5.51 (d, J = 1.8 Hz, 1H)	-0.01	5.56 (br, 1H)	-0.06
3	5.82 (d, J = 2.3 Hz, 1H)	5.82 (d, J = 1.8 Hz, 1H)	0.00	5.87 (br, 1H)	-0.05
4	6.25 (d, J = 8.7 Hz, 2H)	6.29 (d, J = 8.8 Hz, 2H)	-0.04	6.31 (d, J = 8.8 Hz, 2H)	-0.06
5	6.82 (d, J = 8.7 Hz, 2H)	6.83 (d, J = 8.8 Hz, 2H)	-0.01	6.87 (d, J = 8.8 Hz, 2H)	-0.05
6	6.86 (d, J = 2.3 Hz, 1H)	6.90 (dd, J = 2.0, 1.2 Hz, 1H)	-0.04	6.92 (s, 1H)	-0.06
7	6.93 (s, 1H)	6.96 (s, 1H)	-0.03	6.95 (s, 1H)	-0.02
8	7.1 (d, J = 2.3 Hz, 1H)	7.11 (dd, J = 2.0, 1.2 Hz, 1H)	-0.01	7.15 (s, 1H)	-0.05
9	7.15 (dd, J = 8.7, 2.3 Hz, 1H)	7.21 (dd, J = 8.8, 2.0 Hz, 1H)	-0.06	7.23 (dd, J = 8.8, 2.4 Hz, 1H)	-0.08
10	8.14 (d, J = 8.7 Hz, 1H)	8.17 (d, J = 8.8 Hz, 1H)	-0.03	8.21 (d, J = 8.8 Hz, 1H)	-0.07
11	9.46 (s, 1H)	9.39 (d, J = 2.0 Hz, 1H)	0.07	9.41 (d, J = 2.4 Hz, 1H)	0.05

13C NMR					
C	Synthetic Vaticahainol C (Albmiranol B)	Natural Albiraminol B	Δx (ppm)	Natural Vaticahainol C	Δx (ppm)
1	48.9	48.4	0.5	49.0	0.5
2	74.3	73.9	0.4	74.4	0.4
3	96.4	95.9	0.4	96.4	0.4
4	102.2	101.7	0.5	102.3	0.5
5	109.5	109.0	0.5	109.5	0.5
6	111.9	111.3	0.6	112.0	0.6
7	114.5	114.1	0.4	114.5	0.4
8	114.6	114.2	0.4	114.7	0.4
9	114.8	114.3	0.5	114.8	0.5
10	115.5	114.9	0.6	115.1	0.6
11	115.7	115.1	0.6	115.6	0.6
12	115.7	115.2	0.5	115.9	0.5
13	115.9	115.3	0.6	116.1	0.6
14	121.9	121.4	0.5	121.9	0.5
15	130.9	130.5	0.4	130.9	0.4
16	132.8	131.8	1.0	133.4	1.0
17	133.5	133.0	0.5	133.7	0.5
18	133.7	133.3	0.4	133.7	0.4
19	139.6	139.1	0.5	139.6	0.5
20	151.5	151.0	0.5	151.6	0.5
21	155.1	154.7	0.4	155.0	0.4
22	155.8	155.4	0.4	155.8	0.4
23	156.3	155.8	0.4	156.3	0.4
24	156.7	156.4	0.3	156.8	0.3
25	157.1	156.8	0.3	157.2	0.3
26	157.6	157.3	0.3	157.5	0.3

1H NMR			
H	Synthetic Hopeanol A	Natural Hopeanol A	Δx (ppm)
1	6.00 (dd, J = 10.0, 2.1 Hz, 1H)	6.05 (dd, J = 10.0, 1.3 Hz, 1H)	-0.05
2	6.12 (dd, J = 10.3, 2.1 Hz, 1H)	6.17 (dd, J = 10.2, 1.3 Hz, 1H)	-0.05
3	6.45 (dd, J = 8.9, 2.1 Hz, 1H)	6.46 (dd, J = 8.5, 2.1 Hz, 1H)	-0.01
4	6.48 (d, J = 2.1 Hz, 1H)	6.49 (d, J = 1.7 Hz, 1H)	-0.01
5	6.52 (dd, J = 8.9, 2.1 Hz, 1H)	6.53 (dd, J = 8.5, 1.9 Hz, 1H)	-0.01
6	6.66 (dd, J = 8.3, 2.1 Hz, 1H)	6.68 (dd, J = 8.6, 2.1 Hz, 1H)	-0.02
7	6.69 (dd, J = 10.0, 2.8 Hz, 1H)	6.74 (dd, J = 10.0, 2.4 Hz, 1H)	-0.05
8	6.90 (d, J = 2.1 Hz, 1H)	6.90 (d, J = 1.7 Hz, 1H)	0.00
9	7.10 (d, J = 2.1 Hz, 1H)	7.11 (d, J = 2.1 Hz, 1H)	-0.01
10	7.11 (dd, J = 8.3, 2.1 Hz, 1H)	7.12 (dd, J = 8.6, 1.9 Hz, 1H)	-0.01
11	7.37 (dd, J = 10.3, 2.8 Hz, 1H)	7.41 (dd, J = 10.2, 2.4 Hz, 1H)	-0.04
12	7.44 (d, J = 2.1 Hz, 1H)	7.44 (d, J = 2.1 Hz, 1H)	0.00

13C NMR			
C	Synthetic Hopeanol A	Natural Hopeanol A	Δx (ppm)
1	59.2	59.1	0.1
2	102.2	102.1	0.1
3	104.7	104.6	0.1
4	106.1	106.0	0.1
5	110.3	109.9	0.3
6	111.0	110.9	0.1
7	115.0	114.8	0.2
8	117.6	117.5	0.1
9	123.7	123.4	0.3
10	127.2	127.1	0.1
11	129.4	129.1	0.3
12	129.4	129.2	0.2
13	130.4	130.3	0.1
14	130.7	130.7	0.0
15	132.2	132.0	0.2
16	135.3	135.0	0.3
17	136.5	136.8	-0.3
18	138.9	139.3	-0.4
19	142.1	142.0	0.1
20	149.4	150.3	-0.9
21	154.0	153.9	0.1
22	157.9	157.8	0.0
23	158.3	158.3	0.0
24	159.9	160.0	-0.1
25	160.0	160.0	0.0
26	174.8	174.7	0.1
27	186.6	186.9	-0.3
28	187.6	187.6	0.0

REVIEWER COMMENTS

Reviewer #1 (Remarks to the Author):

I am satisfied with the changes made in response to the referees' comments and appreciate the authors' efforts in addressing them thoroughly.

Reviewer #2 (Remarks to the Author):

Regarding the comment D, the authors should clearly state that they compared the gas phase calculated number to the in-solution experimental number at the relevant part of the manuscript (Figure 4 or the relevant paragraph); and optionally add an endnote to show the rationalization of this choice mentioned in the response. Revisions to the other comments are acceptable.

Reviewer #3 (Remarks to the Author):

The manuscript on a superficial level has addressed all of the concerns. However, the changing computational results presented here make me not have confidence in the results. Unfortunately there are too many controls to describe to ensure that the global minima have been found -- and it's not clear that those have been done or that these calculations are valid. The calculations are not properly benchmarked so it is impossible to say if the calculations are sound. I'm not enthusiastic about this manuscript being published.

We thank the reviewers again for helping us to improve our manuscript. All of the authors appreciate the thoughtful and helpful comments. Please find our responses to the reviewers below.

Reviewer #1 (Remarks to the Author):

I am satisfied with the changes made in response to the referees' comments and appreciate the authors' efforts in addressing them thoroughly.

Our response:

Thank you for your comment.

Reviewer #2 (Remarks to the Author):

Regarding the comment D, the authors should clearly state that they compared the gasphase calculated number to the in-solution experimental number at the relevant part of the manuscript (Figure 4 or the relevant paragraph); and optionally add an endnote to show the rationalization of this choice mentioned in the response. Revisions to the other comments are acceptable.

Our response:

We added the following sentence as suggested.

“Although the transition states of acylation were highly charged, we expected that the $\Delta\Delta G$ derived from steric hindrance of the asymmetric catalyst even in the gas phase would be a guideline for choosing an asymmetric catalyst.”

Reviewer #3 (Remarks to the Author):

The manuscript on a superficial level has addressed all of the concerns. However, the changing computational results presented here make me not have confidence in the results. Unfortunately there are too many controls to describe to ensure that the global minima have been found -- and it's not clear that those have been done or that these calculations are valid. The calculations are not properly benchmarked so it is impossible to say if the calculations are sound. I'm not enthusiastic about this manuscript being published.

Our response:

We do not control any results in our conformation searches and benchmarking studies.

Here, we first explain how we had done the conformation search and benchmarking study using Malibatol A.

(1) 73 structures were automatically generated at MM level using Spartan.

(2) All 73 conformers were optimized at ω B97X-D/6-31G** level in H₂O (SMD), and frequency calculations were performed after the geometry optimization at the same level.

(3) The results were sorted with calculated G, and the conformer with the lowest G was determined as the global minimum.

(4) The global minimum at ω B97X-D/6-31G** level in H₂O was used as the initial geometry for the benchmarking study with various functionals and basis sets to optimize the structure.

(5) Using the geometry at ab initio MP2/6-311G** level in H₂O as a reference, the similarity of the structure was compared by RMSD.

(6) ω B97X-D, which had the most similar structure to MP2 calculation, was determined as the optimal functional.

- ① 73 conformers generated by Spartan (MM)
- ② Opt and Freq of all 73 conformers by DFT (ω B97X-D/6-31G** in H₂O) with Gaussian16
- ③ Sort by G and determine a global minimum

Global minimum of Malibatol A (ω B97X-D/6-31G** in H₂O)

- ④ Optimization with various functionals/basis sets using global minimum (ω B97X-D/6-31G** in H₂O) as initial geometry
- ⑤ Compared the optimized structure with the result of MP2/6-311G** in H₂O by RMSD

Method	ab initio (MP2)	DFT												
Functional		B3LYP	B3PW91	B98	BLYP	cam-B3LYP	mPW1PW91	PBEPBE	PBE1PBE	M06	M06-L	M06-2X	wB97	wB97X-D
Basis set	6-311G**	6-31G**												
RMSD (Å)	0.00	0.37	0.35	0.34	0.40	0.34	0.32	0.34	0.31	0.29	0.20	0.19	0.26	0.17

- ⑥ ω B97X-D/6-31G** in H₂O gave the smallest RMSD (0.17).

This reviewer's comments do not expressly point out which data is controlled, but the reviewer may be likely concerned that the conformational search was performed at the ω B97X-D /6-31G** level before the benchmarking study. Therefore, we performed a conformation search with B3LYP / 6-31G** in H₂O and conducted a similar benchmarking study. The global minimum conformation at B3LYP/6-31G** level in H₂O was different from that at ω B97X-D /6-31G** level: one H-O direction of phenol was different. This result is reasonable because different functionals give different results. We then took the benchmarking study. The global minimum at B3LYP/6-31G** level in H₂O was used as the initial geometry for the benchmarking study with various functionals and basis sets to optimize the structure. Then using the geometry at ab initio MP2/6-311G** level in H₂O as a reference, the similarity of the structure was compared by RMSD. The RMSD given by this approach (global minimum at B3LYP/6-31G** level as initial geometry) is almost the same as in the previous calculation (global minimum at ω B97X-D /6-31G** level as initial geometry), and the calculated structure with ω B97X-D /6-31G** is the most similar to that with MP2 calculation.

①
73 conformers generated by Spartan (MM)

②
Opt and Freq of all 73 conformers by DFT (B3LYP/6-31G** in H₂O) with Gaussian16

③
Sort by G and determine a global minimum

Global minimum of Malibatol A (B3LYP/6-31G** in H₂O)

④
Optimization with various functionals/basis sets using global minimum (B3LYP/6-31G** in H₂O) as initial geometry

⑤
Compared the optimized structure with the result of MP2/6-311G** in H₂O by RMSD

Method	ab initio (MP2)	DFT												
Functional		B3LYP	B3PW91	B98	BLYP	cam-B3LYP	mPW1PW91	PBEPBE	PBE1PBE	M06	M06-L	M06-2X	wB97	wB97X-D
Basis set	6-311G**	6-31G**												
RMSD (Å)	0.00	0.37	0.35	0.34	0.40	0.33	0.33	0.34	0.31	0.29	0.20	0.19	0.26	0.17

⑥
wB97X-D/6-31G** in H₂O gave the smallest RMSD (0.17).

In Fig. 2 in this manuscript, TS1 and TS7 are the most important transition states, which support the hypothesis that the natural product epimers epi-Vaticahainol A and B are favorably generated. Therefore, we have performed the transition state calculations again with various functional/basis sets to clear the doubt. With all calculation results, TS1' has lower activation energy (5.1-18.8 kcal/mol) than TS1, and TS7 requires high energy (27.6-37.0 kcal/mol), indicating that no matter which functionals/basis-sets is used, the generation of epi-Vaticahainol B is kinetically favored and the epimerization between epi-Vaticahainol B and Vaticahainol B hardly proceed. Moreover, these computational hypotheses have already been proven by our total synthesis and X-ray analysis. Thus, we think that the benchmarking study for the calculations is sufficient.

We have confidence in our results that we do not control any data. All the geometry, including conformation search and benchmarking study, are provided in the SI, and all original computational output files have been stored.

num	Opt/Freq		TS1	TS1'	
	Functional	Basis set	ΔG	ΔG	
1	wB97X-D	6-31G**	8.7	0.0	
2	wB97		7.7	0.0	
3	B3LYP		6.1	0.0	
4	B98		5.6	0.0	
5	BLYP		5.8	0.0	
6	B3PW91		5.9	0.0	
7	cam-B3LYP		6.7	0.0	
8	mPW1PW91		6.3	0.0	
9	M06		5.1	0.0	
10	M06-L		7.7	0.0	
11	M06-2X		6.9	0.0	
12	PBEPBE		6.1	0.0	
13	PBE1PBE		6.2	0.0	
14	HF		5.4	0.0	
15	wB97X-D		def2SVP	18.8	0.0
16	wB97X-D		cc-pvDZ	7.5	0.0

(kcal/mol)

num	Opt/Freq		PD1	TS7	PD1'	
	Functional	Basis set	ΔG	ΔG	ΔG	
1	wB97X-D	6-31G**	2.0	35.8	0.0	
2	wB97		1.4	34.9	0.0	
3	B3LYP		-0.4	31.9	0.0	
4	B98		0.3	33.5	0.0	
5	BLYP		-0.4	27.6	0.0	
6	B3PW91		-1.1	34.6	0.0	
7	cam-B3LYP		0.7	36.0	0.0	
8	mPW1PW91		-0.1	36.1	0.0	
9	M06		0.8	33.3	0.0	
10	M06-L		0.9	28.9	0.0	
11	M06-2X		1.6	36.1	0.0	
12	PBEPBE		-0.2	32.1	0.0	
13	PBE1PBE		0.1	37.0	0.0	
14	HF		0.5	36.8	0.0	
15	wB97X-D		def2SVP	0.8	34.8	0.0
16	wB97X-D		cc-pvDZ	2.7	35.2	0.0

(kcal/mol)

We have added the information in computational methods section to explain the global minimum shown in this manuscripts (Fig2) is calculated at ω B97X-D /6-31G(d, p) level in water (SMD).

Also, we added the following sentence in the manuscript:

“Moreover, we verified **TS1**, **TS1'** and **TS7** with various functional/basis-set, and all calculation results supported that the generation of C7b-epi-vaticahainol B was kinetically favored and its epimerization hardly proceeded (see Supplementary Fig. 17).”

REVIEWERS' COMMENTS

Reviewer #3 (Remarks to the Author):

The authors additional calculations comparing different functionals is convincing. The central conclusions can be supported and the benchmarking originally found concerning is addressed. The authors have made a stunning finding that is both surprising and informative for future researchers. Congratulations!

We thank the reviewer again for helping us to improve our manuscript. All of the authors appreciate the thoughtful and helpful comments. Please find our responses to the reviewer below.

Reviewer #3 (Remarks to the Author):

The authors additional calculations comparing different functionals is convincing. The central conclusions can be supported and the benchmarking originally found concerning is addressed. The authors have made a stunning finding that is both surprising and informative for future researchers. Congratulations!

We really appreciate your help.